# Genetic Association between Inflammatory-Related Polymorphism in *STAT3*, *IL-1β*, *IL-6*, *TNF-α* and Idiopathic Recurrent Implantation Failure

**DOI:** 10.3390/genes14081588

**Published:** 2023-08-05

**Authors:** Min Jung Kwon, Ji Hyang Kim, Kyu Jae Kim, Eun Ju Ko, Jeong Yong Lee, Chang Su Ryu, Yong Hyun Ha, Young Ran Kim, Nam Keun Kim

**Affiliations:** 1Department of Biomedical Science, College of Life Science, CHA University, Seongnam 13496, Republic of Korea; 0906sally@naver.com (M.J.K.); rbwo0600@naver.com (K.J.K.); ejko05@naver.com (E.J.K.); smilee3625@naver.com (J.Y.L.); regis2040@nate.com (C.S.R.); hayo119@naver.com (Y.H.H.); 2Department of Obstetrics and Gynecology, CHA Bundang Medical Center, School of Medicine, CHA University, Seongnam 13496, Republic of Korea; bin0906@chamc.co.kr

**Keywords:** Recurrent implantation failure (RIF), *STAT3*, *IL-1β*, *IL-6*, *TNF-α*, interleukin (IL), single nucleotide polymorphism (SNP)

## Abstract

Recurrent implantation failure (RIF) is defined as a failure to achieve pregnancy after multiple embryo transfers. Implantation is closely related to inflammatory gradients, and interleukin-1beta (IL-1β), IL-6, and tumor necrosis factor-alpha (TNF-α) play a key role in maternal and trophoblast inflammation during implantation. Signal transducer and activator of transcription 3 (STAT3) interacts with cytokines and plays a critical role in implantation through involvement in the inflammation of the embryo and placenta. Therefore, we investigated 151 RIF patients and 321 healthy controls in Korea and analyzed the association between the polymorphisms (*STAT3* rs1053004, *IL-1β* rs16944, *IL-6* rs1800796, and *TNF-α* rs1800629, 1800630) and RIF prevalence. In this paper, we identified that *STAT3* rs1053004 (AG, adjusted odds rate [AOR] = 0.623; *p* = 0.027; GG, AOR = 0.513; *p* = 0.043; Dominant, AOR = 0.601, *p* = 0.011), *IL-6* rs1800796 (GG, AOR = 2.472; *p* = 0.032; Recessive, AOR = 2.374, *p* = 0.037), and *TNF-α* rs1800629 (GA, AOR = 2.127, *p* = 0.010, Dominant, AOR = 2.198, *p* = 0.007) have a significant association with RIF prevalence. This study is the first to investigate the association of each polymorphism with RIF prevalence in Korea and to compare their effect based on their function on inflammation.

## 1. Introduction

Recurrent implantation failure (RIF) refers to the failure to achieve pregnancy after two or more transfers of high-quality embryos to the endometrium. Among women who suffered from two consecutive pregnancy losses, about 75% are due to an implantation failure [1]. Multiple factors such as the maternal immune system and the genetics of the embryo have been reported to cause RIF, but the mechanism of RIF pathology is still unclear. Implantation is a complex process that requires a competent blastocyst, receptive endometrium, and successful crosstalk between the embryonic and maternal interfaces [2]. Various factors affect the implantation process, including the inflammatory gradients that allow the apposition and adhesion of the blastocyst to the epithelium [3,4]. Therefore, we analyzed the polymorphisms of inflammation genes and investigated associations between the polymorphisms and RIF prevalence to suggest biomarkers for the treatment and diagnosis of RIF.

Diverse pro-inflammatory cytokines and chemokines activate inflammation, and certain cytokines play a key role in maternal and trophoblast inflammation during implantation [5]. Interleukin-1beta (IL-1β), IL-6, and tumor necrosis factor-alpha (TNF-α) are the main pro-inflammatory cytokines secreted by endometrial cells that promote inflammation and affect implantation [5,6]. IL-1β is involved in regulating a cellular adhesion molecule, β3 integrin, for the successful implantation of the blastocyst and has been well-characterized in inflammatory diseases [7]. IL-6, which widely expresses in the female reproductive tract and gestational tissues, plays an important role for a successful pregnancy. IL-6 affects embryo implantation and placental development as well as the immune adaptations required to tolerate pregnancy [8]. TNF-α, which is secreted from the endometrium and plays a critical role in the inflammation process, has been associated with recurrent miscarriage and infertility [7,9]. The pro-inflammatory cytokines IL-1β, IL-6, and TNF-α interact with signal transducer and activator of transcription 3 (STAT3), which is well known to affect embryo implantation and be involved in inflammation. STAT3 has been known to be associated with inflammation-associated tumorigenesis, which is induced by genetic alterations and essential in the mouse uterus during embryo implantation [10,11]. These cytokines and the *STAT3* gene are associated with embryo implantation because of their function in modulating endometrium conditions and have been reported to play a role in preeclampsia and preterm labor [12,13,14].

The genetic factors of RIF have been investigated to uncover its pathology and develop current treatments. In previous papers, several polymorphisms were suggested to be associated with RIF prevalence as well as pregnancy complications [15,16,17,18]. Single nucleotide polymorphisms (SNPs) can contribute to disease pathology through altering the DNA sequence, resulting in a protein with abnormal function [19]. In this study, we investigated prospective SNPs of each inflammation-related gene (*STAT3* rs1053004, rs1053023, *IL-1β* rs16944, *IL-6* rs1800796, and *TNF-α* rs1800629, rs1800630), based on their location in genes, previous clinical papers, and adequate minor allele frequency (MAF). Notably, all polymorphisms investigated in this study are located in gene regulatory regions. *STAT3* rs1053004 A>G and rs1053023 T>C are located in the 3′-untranslated region (UTR), which regulates gene expression [20]; *IL-6* rs1800796 C>G is located in an intron that participates in regulating transcription [21]; and *IL-1β* rs16944 A>G, as well as *TNF-α* rs1800629 G>A and rs1800630 C>A, are located in the promoter, where the transcription of a gene is initiated [22] (Appendix A). In addition, these polymorphisms are associated with pregnancy complications or abnormal inflammatory conditions, which play an important role during implantation. Therefore, we investigated the association between *IL-1β*, *IL-6, TNF-α*, and *STAT3* polymorphisms and RIF prevalence to identify prospective biomarkers for the diagnosis and treatment of RIF. This study is the first, of our knowledge, to analyze the effect of these polymorphisms on RIF prevalence regarding inflammation in the Korean population.

## 2. Materials and Methods

### 2.1. Study Population

RIF patients and controls were collected from the Department of Obstetrics and Gynecology of CHA Bundang Medical Center (Seongnam, Republic of Korea) between March 2010 and December 2012. We collected 151 Korean RIF patients and 321 healthy female Korean controls. The Institutional Review Board of CHA Bundang Medical Center reviewed and approved the study on 23 February 2010 (reference no. CHAMC2009-12-120), and informed consent was obtained from all participants.

RIF was diagnosed when pregnancy failure occurred following the completion of at least two consecutive fresh or frozen IVF-ET cycles using no more than 2 embryos each. All embryos were examined by the embryologist before transfer and only embryos considered to be of good quality were transferred. A good quality embryo is defined by its morphological grade and developmental stage. On day 3 after fertilization, an embryo with at least six blastomeres and a morphological grade of A or B is considered to be of good quality. By day 5, a good-quality embryo is at the 3BB stage or higher in the blastocyst stage [23]. Serum human chorionic gonadotrophin concentration of all samples was less than 5 U/mL, indicating non-pregnancy at 14 days after embryo transfer.

Certain subjects were excluded from the study group when diagnosed with implantation failure due to anatomic, chromosomal, hormonal, infectious, autoimmune, or thrombotic causes, which are exclusion criteria commonly adopted when diagnosing RIF. The anatomical abnormalities of the subjects were evaluated through sonography, hysterosalpingography, hysteroscopy, computed tomography, or magnetic resonance imaging. Chromosomal abnormalities were assessed via karyotyping using standard protocols. Hormonal causes, such as hyperprolactinemia, luteal insufficiency, and thyroid disease, were identified through measuring levels of prolactin, thyroid-stimulating hormone (TSH), free thyroxine, follicle-stimulating hormone (FSH), luteinizing hormone (LH), and progesterone in peripheral blood. Autoimmune diseases including lupus and antiphospholipid syndrome were identified through examining lupus anticoagulant and anticardiolipin antibodies, respectively. Thrombotic causes were defined as thrombophilia and were determined based on deficiencies in protein C and protein S as well as the presence of an anti-β2 glycoprotein antibody. In addition, subjects with a uterine anomaly or underlying medical disease such as endometriosis and polycystic ovarian syndrome were excluded. The control group had regular menstrual cycles, normal karyotype (46XX), no history of pregnancy complications such as pregnancy loss or pre-eclampsia, and at least one natural birth in a healthy condition.

### 2.2. Estimation of Biochemical Factors Concentration

Blood samples were collected from RIF patients after 12 h of fasting. Homocysteine was measured via a fluorescence polarization immunoassay using the Abbott IMx analyzer (Abbott Laboratories, Abbott Park, IL, USA). Folic acid was measured via a competitive immunoassay using the ACS 180Plus automated chemiluminescence system (Bayer Diagnostics, Tarrytown, NY, USA). Total cholesterol, uric acid, blood urea nitrogen (BUN), and creatinine were measured using commercially available enzymatic colorimetric tests (Roche Diagnostics, GmbH, Mannheim, Germany). Platelet, white blood cell, and hemoglobin counts were obtained using the Sysmex XE 2100 automated hematology system (Sysmex Corporation, Kobe, Japan). Prothrombin time and activated partial thromboplastin time (aPTT) were measured with the ACL TOP automated photo-optical coagulometer (LSI Medience, Tokyo, Japan). Hormone levels were assessed after collecting blood samples through venipuncture from women on the second or third day of their menstrual cycles. Serum samples were collected from the blood before measurement. Estradiol, TSH, and prolactin levels were measured using radioimmunoassay (Beckman Coulter, Brea, CA, USA), and FSH and LH levels were measured using enzyme immunoassays (Siemens, Munich, Germany) according to the manufacturer’s instructions.

### 2.3. Genotyping

Genomic DNA was extracted from leukocytes using the G-DEX blood extraction kit (Intron, Seongnam, Republic of Korea). The buffy coat layer containing leukocytes was obtained through collecting the peripheral blood samples and centrifuging them at 3000 rpm for 15 min. The genetic polymorphism of *TNF-α* rs1800630 was analyzed by means of polymerase chain reaction (PCR)-restriction fragment length polymorphism (RFLP) analysis, and all other genetic polymorphisms were analyzed using TaqMan probe real-time PCR (RG–6000, Corbett Research, Mortlake, Australia). *Hpy*CH4IV restriction enzyme (New England Biolabs, Beverly, MA, USA) was used to digest the product for PCR–RFLP analysis. We randomly selected 10% of the PCR assays for each polymorphism and confirmed genotypes through sequencing using an automated sequencer (ABI 3730XL DNA analyzer, Applied Biosystems, Foster City, CA, USA).

### 2.4. Statistical Analysis

The genotype frequencies of each genetic polymorphism were compared between RIF patients and controls using logistic regression. The strength of the association between polymorphisms and RIF was evaluated with an odds ratio (OR) and 95% confidence interval (CI), and the significance of the association was measured using adjusted ORs and 95% CIs adjusted for age from the logistic regression. All polymorphisms were assessed for Hardy–Weinberg equilibrium (HWE) using *p* < 0.05 to investigate deviation and satisfied the HWE. Clinical characteristics were compared between control and patient groups using the chi-square test for categorical data and the two-sided t-test for continuous data. Gene–gene interactions between six genetic loci were assessed using the multifactor dimensionality reduction (MDR) method and MDR software version 2.0 (www.epistasis.org, accessed on 15 February 2023). Statistical analyses were performed using GraphPad Prism 4.0 (GraphPad Software Inc., San Diego, CA, USA), HAPSTAT 3.0 (University of North Carolina, Chapel Hill, NC, USA), and MedCalc version 20.2 (Medcalc Software, Mariakerke, Belgium). *p* values less than 0.05 were considered statistically significant, and the statistical power was calculated via G*power 3 (Appendix A).

## 3. Results

### 3.1. Clinical Profiles of Study Subjects

The clinical characteristics of 321 control subjects and 151 RIF patients are shown in Table 1. There were no significant differences in age for each case–control comparison (*p* = 0.058). Bun, creatinine, uric acid, and prothrombin were significantly higher in the RIF patient group than in the control group (*p* < 0.05). The hormone levels of FSH and LH were not statistically different between the two groups (*p* > 0.05); however, estradiol concentration was significantly higher in the patient group (*p* < 0.0001).

### 3.2. Comparison of Genotype Frequencies of STAT3, IL-1β, IL-6, and TNF-α Polymorphisms

The genotype frequencies of STAT3 rs1053004, IL-1β rs16944, IL-6 rs1800796, and TNF-α rs1800629, and rs1800630 polymorphisms in controls and RIF patients are shown in Table 2. All genetic polymorphisms met the Hardy–Weinberg equilibrium (HWE) (*p* > 0.05). Table 2 shows STAT3 rs1053004 was associated with lower susceptibility to RIF (AG, AOR = 0.623, *p* = 0.027; GG, AOR = 0.513, *p* = 0.043; Dominant, AOR = 0.601, *p* = 0.011). Conversely, IL-6 rs1800796 (GG, AOR = 2.472, *p* = 0.032; Recessive, AOR = 2.374, *p* = 0.037) and TNF-α rs1800629 (GA, AOR = 2.127, *p* = 0.010; Recessive, AOR = 2.198, *p* = 0.007) were significantly associated with the increased occurrence of RIF.

Significant polymorphisms were also associated with RIF occurrence when the patient groups were divided into two groups based on the number of implantation failures (IFs). The RIF patients who suffered greater numbers of IF showed an association between *STAT3* rs1053004 and a lower risk of RIF, and between *IL-6* rs1800796 and *TNF-α* rs1800629 and a higher risk of RIF, revealing a clear correlation between the polymorphisms and IF (Table 3).

### 3.3. Genotype Combination Analysis

Genotype combination analysis was performed to confirm the combined genotype effect of the six SNPs. In genotype combination analysis, the combined genotype of *STAT3* rs1053004/*IL-6* rs1800796, *STAT3* rs1053004/*TNF-α* rs1800629, *STAT3* rs1053004/*STAT3* rs1053023, *STAT3* rs1053023/*IL-6* rs1800796, *IL-6* rs1800796/*TNF-α* rs1800629, and *TNF-α* rs1800629/*TNF-α* rs1800630 showed an association with RIF prevalence (Table 4).

### 3.4. Allele Combination Analysis

Allele combination analysis of the investigated polymorphisms was conducted to analyze the combined effect on RIF occurrence. Prior to allele combination analysis, MDR was used to predict significant interactions between the SNPs. The *STAT3* rs1053004 was selected as the best MDR model and a combination of *STAT3 rs1053004/IL-1β rs16944/IL-6 rs1800796/TNF-α rs1800629, STAT3 rs1053004/IL-6 rs1800796/TNF-α rs1800629,* and *STAT3 rs1053004/TNF-α rs1800629* were predicted to have significance in RIF prevalence. Among the allele combinations, the combination of A-A-G-G of *STAT3* rs1053004/*IL-1β* rs16944/*IL-6* rs1800796/*TNF-α* rs1800629 showed the strongest association with RIF risk (OR = 2.364, 95% CI = 1.090–5.127, *p* = 0.025), but other combinations showed no significance (Table 5).

### 3.5. Clinical Parameter Analysis in Subjects Based on Polymorphisms

Considering the complexity of RIF pathology, an analysis of variance (ANOVA) was performed to analyze the relationship between polymorphisms and various clinical factors. A variety of clinical factors affect RIF, and specific genetic polymorphisms may correlate with specific clinical factors. In Table 6, folate concentration, which is essential in successful implantation [24], was significantly associated with *STAT3* rs1053004. The *STAT3* rs1053004 GG genotype correlated with significantly higher folate levels relative to the AG and AA genotypes. An association between *STAT3* rs1053023 and folate levels as well as BUN was observed. *IL-6* rs1800796 CG genotype is significantly associated with increased platelet levels, which is undesirable during implantation [25].

## 4. Discussion

Previous studies identified genetic factors considered to affect RIF, and several polymorphisms are suggested to be associated with RIF prevalence in our previous papers [18,26,27,28]. In this study, we investigated the association between polymorphisms and RIF prevalence. Among the investigated polymorphisms, *STAT3* rs1053004, *IL-6* rs1800796, and *TNF-α* rs1800629 showed significant effects on RIF prevalence. Considering the resulting proteins’ function in implantation process and inflammation, we suggest *STAT3* rs1053004, *IL-6* rs1800796, and *TNF-α* rs1800629 polymorphisms are functional SNPs which alter expression resulting in different inflammation responses during implantation.

Abnormal inflammation is considered a cause of RIF [29], and elevated levels of inflammation negatively affect sperm and egg quality [30]. *STAT3* acts as an anti-inflammatory factor through activating immune suppressor IL-10 [31,32] while IL-6 and TNF-α are key pro-inflammatory cytokines that mediate the inflammation cascade [33]. Additionally, the IL-1 family is well-known to be involved in inflammation; specifically, the role of IL-1β in relation to the development of inflammatory diseases is well established [34]. Moreover, studies of the *STAT3* rs1053004, *IL-6* rs1800796, and *TNF-α* rs1800629 polymorphisms suggested their significant association with recurrent miscarriage as well as abnormal inflammation responses [6,35,36,37,38,39]. Specifically, the *IL-6* rs1800796 G and *TNF-α* rs1800629 A alleles have significant associations with an increased risk of pregnancy complications [40,41]. In line with previous research on the function of these genes in inflammation, we found that the *STAT3* rs1053004 G allele decreased the risk of RIF, while the *IL-6* rs1800796 G and *TNF-α* rs1800629 A alleles increased the risk of RIF. Therefore, we suggest the *STAT3* rs1053004 G allele suppresses inflammation, resulting in decreased risk of RIF, while the *IL-6* rs1800796 G and *TNF-α* rs1800629 A alleles induce inflammation, resulting in an increased risk of RIF (Figure 1).

Our results further support our hypothesis that *STAT3* rs1053004 and *IL-6* rs1800796 polymorphisms are functional SNPs affecting inflammation and implantation. We found that the *STAT3* rs1053004 GG genotype significantly correlated with increased folate levels while the *IL-6* rs1800796 CG genotype significantly correlated with increased platelet levels. Folate and platelet levels are important clinical factors for implantation and inflammation. Folate is involved in inflammatory diseases and is vital during pregnancy and infancy [42,43]. Low folate levels reduce growth rates and increase the expression of inflammatory mediators, which can induce a number of pregnancy-related complications [44]. Thus, we suggest that the *STAT3* rs1053004 GG genotype may suppress inflammation through increasing folate levels as a mechanism for decreasing the risk of RIF. Platelet levels are also important during pregnancy and the development of inflammation [45,46]. Specifically, activated platelets promote inflammatory diseases, and platelet levels normally decrease during pregnancy. This finding supports the hypothesis that the *IL-6* rs1800796 CG genotype increases the risk of RIF through inducing inflammation.

However, there are some limitations to our study. First, further studies are necessary to determine the mechanisms responsible for the observed polymorphism associations. Although the studied polymorphisms are located in the regulatory region, the mechanism of gene expression in regulation remains unclear. Further, to clarify the association between polymorphisms and RIF, the correction of various confounding factors, such as other patient characteristics and IVF outcomes, is necessary. However, we used logistic regression for our statistical results and adjusted the analysis by age to address this issue. Lastly, our study population was limited to Korean women, and further research in a more diverse patient cohort is required. Moreover, because RIF is a relatively uncommon disease, our patient sample size was small; more samples will be needed to substantiate our observations.

Elevated inflammation induces RIF; *STAT3* suppresses inflammation; and IL-1β, IL-6, and *TNF-α* induce inflammation. We found that the *STAT3* rs1053004 G allele is associated with decreased risk of RIF, and the *IL-6* rs1800796 G and *TNF-α* rs1800629 A alleles are associated with an increased risk of RIF. In addition, we found significant correlations between *STAT3* rs1053004 polymorphism and folate levels as well as *IL-6* rs1800796 and platelet levels. Based on our results, we hypothesized that each SNP affects gene expression, which alters the inflammatory response and RIF occurrence.

## 5. Conclusions

In this study, we analyzed the association between genetic polymorphisms involved in the *STAT3* signaling pathway and RIF prevalence in Korean women. Based on the fact that STAT3 has an important role in inflammation, we investigated genetic polymorphisms of well-known pro-inflammatory genes. As a result, the *STAT3* rs1053004 G allele showed a significant association with decreased risk of RIF, while the *IL-6* rs1800796 G and *TNF-α* rs1800629 G alleles showed a significant association with increased risk of RIF. In addition, the genetic associations increased in significance as the number of implantation failures increased, confirming the association between the SNPs and implantation failure. Furthermore, we identified a significant correlation between inflammation-related factors and the *STAT3* rs1053004 GG genotype and the *IL-6* rs1800796 CG genotype, respectively. In line with previous studies, our results showed that *STAT3* rs1053004, *IL-6* rs1800796, and *TNF-α* rs1800629 polymorphisms are involved in abnormal inflammation response while affecting RIF prevalence. Although further in vitro and in vivo studies are needed to determine the mechanism of these SNPs, our results indicate that *STAT3* rs1053004, *IL-6* rs1800796, and *TNF-α* rs1800629 polymorphisms, which are located in gene regulatory regions, contribute to abnormal gene function. Therefore, we suggest that they could be the clinical references for personalized treatment of RIF serving as biomarkers.

## Figures and Tables

**Figure 1 genes-14-01588-f001:**
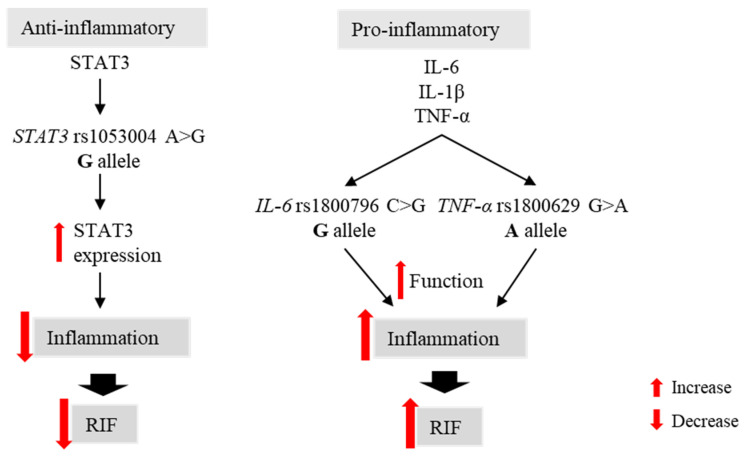
Model of *STAT3*, *IL-1β*, *IL-6*, and *TNF-α* polymorphism function in inflammation and recurrent implantation failure.

**Table 1 genes-14-01588-t001:** Baseline characteristics between patients and control subjects.

Characteristic	Control (n = 321)	RIF (n = 151)	*p*
Age (year, mean ± SD)	33.31 ± 3.46	33.93 ± 3.11	0.058
BMI (kg/m^2^, mean ± SD)	21.89 ± 3.39 (79)	21.33 ± 3.28 (150)	0.175
Previous implantation failure (n, mean ± SD)	None	5.13 ± 2.30 (150)	
Live birth (n, mean ± SD)	1.46 ± 0.51 (22)	None	
Mean gestational age (week, mean ± SD)	39.00 ± 1.53 (18)	None	
BUN (mg/dL, mean ± SD)	8.78 ± 2.81 (194)	10.49 ± 2.85 (105)	<0.0001
Creatinine (mg/dL, mean ± SD)	0.64 ± 0.16 (193)	0.78 ± 0.10 (106)	<0.0001
Hcy (μmol/L, mean ± SD)	6.28 ± 2.90 (24)	6.79 ± 1.74 (64)	0.273
Folate (ng/mL, mean ± SD)	15.76 ± 10.40 (24)	14.26 ± 7.78 (44)	0.724
Uric acid (mg/dL, mean ± SD)	3.79 ± 1.03 (136)	4.06 ± 0.98 (70)	0.046
PLT (103/μL, mean ± SD)	2.42 ± 0.60 (163)	2.44 ± 0.69 (135)	0.932
PT (s, mean ± SD)	10.70 ± 1.56 (199)	10.90 ± 2.04 (134)	<0.0001
aPTT (s, mean ± SD)	29.24 ± 3.56 (199)	29.68 ± 3.50 (134)	0.172
FSH (U/L, mean ± SD)	7.08 ± 1.10 (9)	9.09 ± 4.85 (103)	0.215
LH (U/L, mean ± SD)	4.83 ± 1.87 (8)	5.09 ± 2.52 (103)	0.968
E2 (Estradiol) (pg/mL, mean ± SD)	26.11 ± 14.69 (101)	35.50 ± 16.87 (117)	<0.0001

Note: RIF, recurrent implantation failure; SD, standard deviation; ND, not done; BMI, body mass index; PT, prothrombin; aPTT, activated partial thromboplastin time; Hcy, homocysteine; Hgb, hemoglobin; PLT, platelet count; BUN, blood urea nitrogen; FSH, follicle stimulating hormone; LH, luteinizing hormone; E2, estradiol.

**Table 2 genes-14-01588-t002:** Comparison of genotype frequencies and AOR values of polymorphisms between the RIF patients and control subjects.

Genotypes	Controls (n = 321)	RIF (n = 151)	AOR (95% CI)	*p*	FDR*-P*
*STAT3* rs1053004 A>G					
AA	127 (39.6)	78 (51.7)	1.000 (reference)		
AG	146 (45.5)	58 (38.4)	0.623 (0.409–0.947)	0.027	0.081
GG	48 (15.0)	15 (9.9)	0.513 (0.269–0.978)	0.043	0.108
Dominant (AA vs. AG+GG)			0.601 (0.406–0.889)	0.011	0.033
Recessive (AA+AG vs. GG)			0.642 (0.346–1.190)	0.159	0.199
HWE*-P*	0.570	0.390			
*STAT3* rs1053023 T>C					
TT	166 (51.7)	83 (55.0)	1.000 (reference)		
TC	124 (38.6)	60 (39.7)	0.969 (0.645–1.456)	0.881	0.881
CC	31 (9.7)	8 (5.3)	0.535 (0.235–1.220)	0.137	0.171
Dominant (TT vs. TC+CC)			0.883 (0.598–1.304)	0.532	0.638
Recessive (TT+TC vs. CC)			0.539 (0.241–1.206)	0.133	0.199
HWE*-P*	0.271	0.499			
*IL-1β* rs16944 A>G					
AA	91 (28.3)	35 (23.2)	1.000 (reference)		
AG	155 (48.3)	79 (52.3)	1.217 (0.724–0.0449)	0.459	0.881
GG	75 (23.4)	37 (24.5)	1.165 (0.632–0.1459)	0.624	0.624
Dominant (AA vs. AG+GG)			1.197 (0.731–0.9601)	0.474	0.638
Recessive (AA+AG vs. GG)			1.017 (0.619–0.6686)	0.947	0.947
HWE*-P*	0.568	0.567			
*IL-6* rs1800796 C>G					
CC	196 (61.1)	86 (57.0)	1.000 (reference)		
CG	113 (35.2)	52 (34.4)	1.038 (0.684–1.574)	0.861	0.881
GG	12 (3.7)	13 (8.6)	2.472 (1.083–5.645)	0.032	0.108
Dominant (CC vs. CG+GG)			1.170 (0.789–1.735)	0.436	0.638
Recessive (CC+CG vs. GG)			2.374 (1.053–5.350)	0.037	0.037
HWE *P*	0.384	0.214			
*TNF-α* rs1800629 G>A					
GG	292 (91.0)	124 (82.1)	1.000 (reference)		
GA	29 (9.0)	26 (17.2)	2.127 (1.200–3.769)	0.010	0.060
AA	0 (0.0)	1 (0.7)	N/A		
Dominant (GG vs. GA+AA)			2.198 (1.246–3.878)	0.007	0.033
Recessive (GG+GA vs. AA)			N/A		
HWE*-P*	0.397	0.773			
*TNF-α* rs1800630 C>A					
CC	223 (69.5)	106 (70.2)	1.000 (reference)		
CA	89 (27.4)	44 (29.1)	1.069 (0.694–1.646)	0.762	0.881
AA	9 (3.1)	1 (0.7)	0.206 (0.026–1.636)	0.135	0.171
Dominant (CC vs. CA+AA)			0.978 (0.640–1.494)	0.919	0.919
Recessive (CC+CA vs. AA)			0.203 (0.026–1.603)	0.131	0.131
HWE*-P*	0.715	0.115			

RIF, recurrent implantation failure; AOR, adjusted odds ratio; 95% CI, 95% confidence interval; HWE, Hardy–Weinberg equilibrium; N/A, not applicable; FDR, false discovery rate. AOR is adjusted by age.

**Table 3 genes-14-01588-t003:** Comparison of genotype frequencies and AOR values of polymorphisms between the case and control according to the number of implantation failure (IF).

Genotypes	IF ≥ 3 (n = 105)	AOR (95% CI)	*p*	FDR*-P*	IF ≥ 4 (n = 75)	AOR (95% CI)	*p*	FDR*-P*
*STAT3* rs1053004 A>G								
AA	61 (58.1)	1.000 (reference)			46 (61.1)	1.000 (reference)		
AG	36 (34.1)	0.600 (0.389–0.925)	0.021	0.063	24 (32.1)	0.559 (0.347–0.902)	0.017	0.051
GG	8 (8.1)	0.440 (0.219–0.884)	0.021	0.088	5 (7.1)	0.416 (0.191–0.905)	0.027	0.073
Dominant (AA vs. AG+GG)		0.567 (0.378–0.851)	0.006	0.021		0.533 (0.341–0.833)	0.006	0.018
Recessive (AA+AG vs. GG)		0.559 (0.286–1.093)	0.089	0.148		0.542 (0.256–1.151)	0.111	0.185
*STAT3* rs1053023 T>C								
TT	76 (55.1)	1.000 (reference)			60 (56.1)	1.000 (reference)		
TC	56 (40.6)	0.988 (0.650–1.502)	0.955	0.955	43 (40.2)	0.958 (0.605–1.517)	0.855	0.855
CC	6 (4.3)	0.446 (0.178–1.119)	0.086	0.143	4 (3.7)	0.382 (0.129–1.135)	0.083	0.138
Dominant (TT vs. TC+CC)		0.882 (0.589–1.320)	0.54	0.648		0.847 (0.544–1.320)	0.463	0.463
Recessive (TT+TC vs. CC)		0.445 (0.181–1.097)	0.079	0.148		0.388 (0.133–1.129)	0.082	0.185
*IL–1β* rs16944 A>G								
AA	20 (19.1)	1.000 (reference)			15 (20.1)	1.000 (reference)		
AG	63 (60.1)	1.630 (0.915–0.9023)	0.097	0.194	46 (61.1)	1.588 (0.830–0.0348)	0.162	0.324
GG	22 (21.1)	1.259 (0.628–0.5234)	0.516	0.516	14 (19.1)	1.069 (0.478–0.3903)	0.87	0.870
Dominant (AA vs. AG+GG)		1.510 (0.866–0.6331)	0.146	0.292		1.422 (0.760–0.6604)	0.27	0.405
Recessive (AA+AG vs. GG)		0.877 (0.507–0.5147)	0.637	0.637		0.763 (0.400–0.4547)	0.412	0.412
*IL–6* rs1800796 C>G								
CC	59 (56.1)	1.000 (reference)			42 (56.1)	1.000 (reference)		
CG	36 (34.1)	1.059 (0.689–1.629)	0.794	0.953	25 (33.1)	1.064 (0.663–1.710)	0.797	0.855
GG	10 (10.1)	2.483 (1.068–5.770)	0.035	0.088	8 (11.1)	2.687 (1.104–6.538)	0.029	0.073
Dominant (CC vs. CG+GG)		1.194 (0.794–1.793)	0.395	0.593		1.219 (0.780–1.905)	0.384	0.461
Recessive (CC+CG vs. GG)		2.391 (1.041–5.492)	0.040	0.148		2.594 (1.079–6.237)	0.033	0.165
*TNF-α* rs1800629 G>A								
GG	85 (81.1)	1.000 (reference)			59 (79.1)	1.000 (reference)		
GA	19 (18.1)	2.163 (1.202–3.893)	0.010	0.060	15 (20.1)	2.373 (1.271–4.431)	0.007	0.042
AA	1 (1.1)	N/A			1 (1.1)	N/A		
Dominant (GG vs. GA+AA)		2.239 (1.251–4.008)	0.007	0.021		2.472 (1.333–4.585)	0.004	0.018
Recessive (GG+GA vs. AA)		N/A				N/A		
*TNF-α* rs1800630 C>A								
CC	75 (71.1)	1.000 (reference)			75 (100.1)	1.000 (reference)		
CA	31 (30.1)	1.147 (0.737–1.786)	0.543	0.815	31 (41.1)	1.092 (0.668–1.783)	0.727	0.855
AA	1 (1.1)	0.231 (0.029–1.836)	0.166	0.208	1 (1.1)	0.297 (0.037–2.367)	0.252	0.315
Dominant (CC vs. CA+AA)		1.050 (0.680–1.621)	0.826	0.826		1.005 (0.621–1.627)	0.983	0.405
Recessive (CC+CA vs. AA)		0.222 (0.028–1.757)	0.154	0.193		0.288 (0.036–2.282)	0.239	0.299

IF, implantation failure; AOR, adjusted odds ratio; 95% CI, 95% confidence interval; HWE, Hardy–Weinberg equilibrium; N/A, not applicable; FDR, false discovery rate. AOR is adjusted by age.

**Table 4 genes-14-01588-t004:** Genotype combinations of genetic polymorphisms between controls and RIF patients.

Characteristics	Controls (n = 321)	RIF (n = 151)	AOR (95% CI)	*p*
*STAT3* rs1053004/*IL-6* rs1800796				
AA/CC	79 (24.6)	42 (27.8)	1.000 (reference)	
AA/GG	4 (1.2)	8 (5.3)	3.767 (1.068–13.281)	0.039
*STAT3* rs1053004/*TNF-a* rs1800629				
AA/GG	117 (36.4)	67 (44.4)	1.000 (reference)	
AG/GG	132 (41.1)	49 (32.5)	0.623 (0.397–0.977)	0.039
GG/GG	43 (13.4)	8 (5.3)	0.328 (0.145–0.740)	0.007
*STAT3* rs1053004/*STAT3* rs1053023				
AA/TT	118 (36.8)	71 (47.0)	1.000 (reference)	
AG/TT	44 (13.7)	9 (6.0)	0.326 (0.149–0.712)	0.005
*STAT3* rs1053023/*IL-6* rs1800796				
TT/CC	101 (31.5)	48 (31.8)	1.000 (reference)	
CC/CC	23 (7.2)	3 (2.0)	0.263 (0.074–0.925)	0.037
*STAT3* rs1053023/*TNF-α* rs1800629				
TT/GG	153 (47.7)	70 (46.4)	1.000 (reference)	
CC/GG	28 (8.7)	3 (2.0)	0.244 (0.071–0.832)	0.024
*IL-6* rs1800796/*TNF-α* rs1800629				
CC/GG	179 (55.8)	73 (48.3)	1.000 (reference)	
GG/GA	1 (0.3)	4 (2.6)	9.907 (1.086–90.332)	0.042
*TNF-α* rs1800629/*TNF-α* rs1800630				
GG/CC	197 (61.4)	86 (57.0)	1.000 (reference)	
GA/CA	3 (0.9)	7 (4.6)	5.752 (1.432–3.093)	0.014

RIF, recurrent implantation failure; AOR, adjusted odds ratio; 95% CI, 95% confidence interval. AOR adjusted by age.

**Table 5 genes-14-01588-t005:** MDR-based allele combination analysis of polymorphisms between controls and RIF patients.

Characteristics	Controls (n = 642)	RIF (n = 302)	OR (95% CI)	*p*
*STAT3* rs1053004*/IL-1β* rs16944/ *IL-6* rs1800796*/TNF-a* rs1800629				
A-A-C-G	417 (64.9)	190 (63.0)	1.000 (reference)	
A-A-C-A	2 (0.3)	3 (0.9)	3.292 (0.545–19.870)	0.183
A-A-G-G	13 (2.1)	14 (4.6)	2.364 (1.090–5.127)	0.025
A-A-G-A	0 (0.0)	1 (0.4)	6.575 (0.266–162.300)	0.314
A-G-C-G	68 (10.6)	36 (12.0)	1.162 (0.749–1.802)	0.503
A-G-C-A	2 (0.3)	1 (0.4)	1.097 (0.098–12.180)	1.000
A-G-G-G	17 (2.7)	11 (3.7)	1.420 (0.652–3.091)	0.375
A-G-G-A	2 (0.3)	2 (0.6)	2.195 (0.306–15.710)	0.594
G-A-C-G	46 (7.2)	14 (4.6)	0.668 (0.358–1.245)	0.201
G-A-C-A	2 (0.3)	1 (0.2)	1.097 (0.098–12.180)	1.000
G-A-G-G	10 (1.6)	2 (0.7)	0.439 (0.095–2.024)	0.278
G-A-G-A	0 (0.0)	1 (0.4)	6.575 (0.266–162.300)	0.314
G-G-C-G	34 (5.3)	15 (5.0)	0.968 (0.515–1.821)	0.920
G-G-C-A	3 (0.5)	3 (1.1)	2.195 (0.438–10.980)	0.326
G-G-G-G	23 (3.5)	6 (1.9)	0.573 (0.229–1.429)	0.227
G-G-G-A	3 (0.5)	2 (0.7)	1.463 (0.242–8.833)	0.651
*STAT3* rs1053004*/IL-6* rs1800796/ *TNF-a* rs1800629				
A-C-G	462 (71.9)	204 (67.5)	1.000 (reference)	
A-C-A	4 (0.6)	3 (1.1)	1.699 (0.376–7.661)	0.445
A-G-G	24 (3.8)	16 (5.3)	1.510 (0.785–2.903)	0.214
A-G-A	0 (0.0)	2 (0.8)	11.310 (0.540–236.800)	0.095
G-C-G	103 (16.0)	51 (17.0)	1.121 (0.771–1.630)	0.548
G-C-A	6 (0.9)	5 (1.5)	1.887 (0.569–6.256)	0.291
G-G-G	39 (6.1)	17 (5.6)	0.987 (0.545–1.786)	0.966
G-G-A	5 (0.8)	4 (1.3)	1.812 (0.481–6.819)	0.468
*STAT3* rs1053004/*TNF-a* rs1800629				
A-G	515 (80.2)	251 (83.1)	1.000 (reference)	
A-A	6 (1.0)	7 (2.3)	2.394 (0.795–7.199)	0.109
G-G	113 (17.6)	37 (12.3)	0.672 (0.450–1.003)	0.051
G-A	8 (1.3)	7 (2.3)	1.795 (0.643–5.008)	0.257

MDR, multifactor dimensionality reduction; RIF, recurrent implantation failure; OR, odds ratio; 95% CI, 95% confidence interval.

**Table 6 genes-14-01588-t006:** Differences in various clinical parameters in RIF and control subjects.

Genotypes	PT (SEC)	aPTT	Hcy	Folate	WBC	Hemoglobin	Platelet	BUN	Creatinine	Uric Acid	Total Cholesterol
	Mean ± SD	Mean ± SD	Mean ± SD	Mean ± SD	Mean ± SD	Mean ± SD	Mean ± SD	Mean ± SD	Mean ± SD	Mean ± SD	Mean ± SD
*STAT3* rs1053004 A>G											
AA	10.86 ± 1.13	29.36 ± 3.73	6.40 ± 2.20	15.17 ± 7.76	7.58 ± 2.38	12.38 ± 1.34	238.35 ± 63.82	9.31 ± 2.62	0.71 ± 0.14	3.87 ± 1.02	202.35 ± 56.27
AG	10.76 ± 2.26	29.49 ± 3.44	6.87 ± 2.17	12.60 ± 7.02	7.65 ± 2.69	12.28 ± 1.23	246.68 ± 66.47	9.70 ± 3.20	0.67 ± 0.16	3.93 ± 1.00	202.03 ± 52.39
GG	10.59 ± 1.75	29.40 ± 3.20	6.36 ± 1.33	22.55 ± 13.85	7.84 ± 2.08	12.25 ± 1.27	247.60 ± 60.01	8.48 ± 2.90	0.68 ± 0.18	3.75 ± 1.09	207.15 ± 59.39
*p*	0.679	0.957	0.572	0.041	0.848	0.769	0.519	0.085	0.168	0.740	0.888
*STAT3* rs1053023 T>C											
TT	10.94 ± 0.99	29.51 ± 3.47	6.63 ± 1.66	31.86 ± 13.58	6.72 ± 2.16	12.39 ± 1.20	237.44 ± 63.95	8.24 ± 3.26	0.67 ± 0.17	3.51 ± 0.90	192.26 ± 45.88
TC	10.84 ± 2.38	29.50 ± 3.40	6.76 ± 2.15	12.88 ± 6.38	7.11 ± 2.82	12.63 ± 2.61	253.19 ± 64.21	9.81 ± 3.09	0.69 ± 0.16	3.87 ± 1.02	203.10 ± 56.74
CC	10.72 ± 1.30	29.35 ± 3.65	6.52 ± 2.15	14.62 ± 8.14	7.07 ± 2.38	12.42 ± 1.21	236.04 ± 63.83	9.25 ± 2.73	0.69 ± 0.15	3.95 ± 1.03	204.13 ± 54.90
*p*	0.715	0.823	0.726	0.016	0.717	0.921	0.062	0.036	0.837	0.329	0.617
*IL-1β* rs16944 A>G											
AA	11.04 ± 2.16	29.23 ± 3.52	7.01 ± 2.69	17.31 ± 14.07	7.46 ± 2.67	12.33 ± 1.22	236.92 ± 62.39	9.30 ± 3.05	0.70 ± 0.15	4.08 ± 1.09	202.71 ± 57.89
AG	10.74 ± 1.37	29.42 ± 3.64	6.74 ± 1.69	13.08 ± 6.55	7.66 ± 2.41	12.50 ± 1.17	247.20 ± 64.98	9.67 ± 2.83	0.69 ± 0.16	3.89 ± 1.02	200.32 ± 50.10
GG	10.61 ± 2.00	29.60 ± 3.38	6.31 ± 2.34	16.40 ± 8.41	7.80 ± 2.42	11.98 ± 1.46	241.20 ± 65.48	8.93 ± 2.98	0.67 ± 0.16	3.73 ± 0.95	207.20 ± 60.24
*p*	0.252	0.784	0.519	0.226	0.676	0.056	0.491	0.195	0.530	0.167	0.684
*IL-6* rs1800796 C>G											
CC	10.68 ± 1.62	29.23 ± 3.51	6.74 ± 1.70	15.43 ± 9.69	7.42 ± 2.42	12.30 ± 1.28	233.66 ± 57.20	9.47 ± 2.93	0.70 ± 0.15	3.89 ± 1.05	199.10 ± 56.21
CG	10.86 ± 2.12	29.49 ± 3.64	6.75 ± 2.68	13.63 ± 6.57	7.89 ± 2.49	12.46 ± 1.21	256.51 ± 73.54	9.03 ± 2.77	0.67 ± 0.16	3.81 ± 1.03	207.89 ± 52.85
GG	11.28 ± 0.89	30.73 ± 2.98	5.54 ± 0.83	15.97 ± 11.90	8.50 ± 2.81	11.67 ± 1.56	247.50 ± 53.60	10.33 ± 3.63	0.73 ± 0.15	4.15 ± 0.74	210.90 ± 51.53
*p*	0.276	0.163	0.357	0.692	0.091	0.084	0.009	0.168	0.209	0.462	0.373
*TNF-α* rs1800629 G>A											
GG	10.75 ± 1.88	29.42 ± 3.48	6.63 ± 2.23	15.27 ± 9.10	7.69 ± 2.46	12.34 ± 1.25	243.67 ± 64.94	9.33 ± 2.93	0.68 ± 0.16	3.86 ± 1.02	202.58 ± 55.55
GA	10.95 ± 0.82	29.42 ± 3.93	6.78 ± 1.16	12.56 ± 6.73	7.31 ± 2.64	12.19 ± 1.47	236.73 ± 60.46	9.67 ± 2.97	0.71 ± 0.15	4.08 ± 0.97	204.17 ± 50.23
AA	11.20 ± 0.00	28.30 ± 0.00	N/A	N/A	N/A	N/A	302.00 ± 0.00	11.50 ± 0.00	0.80 ± 0.00	N/A	N/A
*p*	0.783	0.951	0.547	0.330	0.283	0.674	0.545	0.616	0.479	0.313	0.788
*TNF-α* rs1800630 C>A											
CC	10.76 ± 2.08	29.50 ± 3.49	6.73 ± 2.24	15.48 ± 9.14	7.59 ± 2.53	12.23 ± 1.35	241.73 ± 64.18	9.34 ± 2.74	0.69 ± 0.16	3.88 ± 1.00	203.62 ± 56.93
CA	10.88 ± 0.80	29.39 ± 3.69	6.44 ± 1.77	12.88 ± 7.45	7.77 ± 2.42	12.48 ± 1.10	242.77 ± 62.10	9.49 ± 3.26	0.68 ± 0.15	3.89 ± 1.08	201.16 ± 51.27
AA	10.20 ± 0.45	27.31 ± 2.24	N/A	N/A	7.73 ± 1.96	13.00 ± 1.09	300.20 ± 94.01	9.29 ± 4.17	0.70 ± 0.12	4.04 ± 0.81	200.29 ± 45.84
*p*	0.059	0.233	0.912	0.251	0.721	0.208	0.347	0.972	0.934	0.854	0.998

*p* values were calculated via ANOVA. Abbreviations: RIF, recurrent implantation failure; SD, standard deviation; PT, prothrombin; aPTT, activated partial thromboplastin time; Hcy, homocysteine; WBC, white blood cell; BUN, blood urea nitrogen.

## Data Availability

The data presented in this study are openly available in FigShare at 10.6084/m9.figshare.22776083.

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
