# Peer review of "Genetic Association between Inflammatory-Related Polymorphism in STAT3, IL-1β, IL-6, TNF-α and Idiopathic Recurrent Implantation Failure"

_genes, 2023, doi:10.3390/genes14081588_

Round 1

Reviewer 1 Report (New Reviewer)

The authors of the manuscript "Genetic association between .........implantation failure" have provided a well written study and a concise report. Nevertheless, there is a major point which to my opinion has to be revised. The point is that in the 2.1. Study population section line 96 "considered to be of good quality embryos were transferred" a definition of good quality embryos according to the standards the local embryologists used should be added, and the manuscript revised accordingly.

No comment

Author Response

Thank you for your comments. We agree that it will be more understandable if we indicate what is good-quality embryos. Therefore, we inserted the below sentences in 2.1. Study population section.

“A good quality embryo is defined by its morphological grade and developmental stage. On day 3 after fertilization, an embryo with at least six blastomeres and a morphological grade of A or B is considered to be of good quality. By day 5, a good-quality embryo is at the 3BB stage or higher in the blastocyst stage [23].” [Line 96, (page 3, line 1 to 5)]

Reviewer 2 Report (New Reviewer)

The authors of "Genetic association between ............." have designed and implemented a study on the genetic aspects of implantation failure.  The study is sound and well presented.

No comment

The authors of "Genetic association between ............." have designed and implemented a study on the genetic aspects of implantation failure.  The study is sound and well presented and merits publication.

Author Response

Thank you for your comments. I appreciate that you spend your time reviewing our manuscript.

Reviewer 3 Report (New Reviewer)

The authors have made a good attempt at linking polymorphisms in specific inflammatory mediators to RIF.

It is good that the authors noted that there are no age differences  between women with RIF and normal women. However it will be ideal if the authors can include more information on how "good embryo" were defined and if there are any male factor Infertility as we know increased DNA fragmentation can lead to increased risk of recurrent miscarriages and RIF.

Furthermore the association between STAT3/IL6/TNF alpha is a bit confusing- can the authors clarify about why they perform the analyses and how this will impact on diagnosis when using all 3 polymorphisms as markers?

English is relatively accurate 

Author Response

Thank you for your comments. We agree that it will be more understandable if we indicate what is good-quality embryos. Therefore, we inserted the below sentences in 2.1. Study population section. In addition, we agree that sperm DNA fragmentation can affect implantation failure but we don't have data about sperm DNA fragmentation because we do not test it in our clinical practice.

“A good quality embryo is defined by its morphological grade and developmental stage. On day 3 after fertilization, an embryo with at least six blastomeres and a morphological grade of A or B is considered to be of good quality. By day 5, a good-quality embryo is at the 3BB stage or higher in the blastocyst stage [23].” [Line 96, (page 3, line 1 to 5)]

Thank you for your comments. As you suggested, we added some sentences in the Conclusion to reduce the possible confusion among authors.

“Based on the fact that STAT3 has an important role in inflammation, we investigated genetic polymorphisms of well-known pro-inflammatory genes.” [Conclusion, Line 63]

“Although further in vitro and in vivo studies are needed to determine the mechanism of these SNPs, our results indicate that STAT3 rs1053004, IL-6 rs1800796, and TNF-α rs1800629 polymorphisms, which are located in gene regulatory regions, contribute to abnormal gene function. Therefore, we suggest that they could be the clinical references for personalized treatment of RIF serving as biomarkers.” [Conclusion, Line 75]

This manuscript is a resubmission of an earlier submission. The following is a list of the peer review reports and author responses from that submission.

Round 1

Reviewer 1 Report

The main concern:

There is no universal definition of RIF, and women defined as RIF patients represent a very heterogeneous group. Therefore, examining the frequencies of SNP polymorphisms between 151 RIF patients and 321 controls cannot provide reliable results. Further dividing patients according to the number of implantation failures may make the groups more uniform, but further reduces study power.

In addition:

The study group is poorly defined. The cause of infertility must be taken into account. Why did the patients undergo IVF? Were endometriosis and PCOS excluded? Were the patients nulliparous?

RIF was diagnosed when pregnancy failure occurred following the completion of two fresh IVF-ET cycles using more than ten cleaved embryos. This sentence needs clarification, what were ten or more embryos used for - for two IVF-ET cycles? How many embryos were transferred during the cycle?

Reviewer 2 Report

Corrections

line 167 - E2 is significantly higher (not different)

line 168 Table 1 - lists several cell types of CD19-CD8; there is no description in methods of how these measurements were made, correct or remove the data.

line 178 - Stat 3 'was' associated with ...

line 179 - how is Dominant polymorphism determined 

line 180 - IL6 SNP 'GG' is significant not 'CG'

section 3.5 line 35, ...which is 'undesirable' during implantation.

Discussion line 59 - remove consistent

line 60 - each SNP affects 'gene' expression 

Reviewer 3 Report

The choice of polymorphisms studied in the context of recurrent implantation failure (RIF) is logical and well grounded in the pathogenesis of RIF. The summary provides all relevant information. The introduction should be improved: The polymorphisms studied should be described in more detail, since the rs numbers provided (lines 65-83) do not inform the reader about the SNP localization within the gene (promoter? intron? UTR?) and the nucleotide change that occurred. The description of the method is thorough, but the information is  missingwhether the numbers of patients and controls were determined a priori in relation to the desired statistical power? If yes, please specify the instrument used for power calculation.